# Extracranial Germ Cell Tumors in Children: Ten Years of Experience in Three Children’s Medical Centers in Shanghai

**DOI:** 10.3390/cancers15225412

**Published:** 2023-11-14

**Authors:** Shayi Jiang, Kuiran Dong, Kai Li, Jiangbin Liu, Xin Du, Can Huang, Yangyang Jiao, Yali Han, Jingwei Yang, Xuelian Liao, Yanhua Li, Ting Zhang, Shanshan Li, Zhibao Lv, Yijin Gao

**Affiliations:** 1Department of Hematology and Oncology, Shanghai Children’s Hospital, School of Medicine, Shanghai Jiao Tong University, No. 355 Luding Road, Shanghai 200061, China; jiangsy@shchildren.com.cn (S.J.); dux@shchildren.com.cn (X.D.); huangcan@shchildren.com.cn (C.H.); jiaojj@shchildren.com.cn (Y.J.); yangjw@shchildren.com.cn (J.Y.); liaoxl@shchildren.com.cn (X.L.); liyh@shchildren.com.cn (Y.L.); zhangt@shchildren.com.cn (T.Z.); liss@shchildren.com.cn (S.L.); 2Department of Pediatric Surgery, Children’s Hospital of Fudan University, Shanghai 201102, China; kuirand@fudan.edu.cn (K.D.); likai2727@163.com (K.L.); 3Department of Pediatric Surgery, Shanghai Children’s Hospital, School of Medicine, Shanghai Jiao Tong University, No. 355 Luding Road, Shanghai 200061, China; liujb@shchildren.com.cn; 4Department of Hematology and Oncology, Shanghai Children’s Medical Center, School of Medicine, Shanghai Jiao Tong University, No. 1678 Dongfang Road, Shanghai 200127, China; hanyali@scmc.com.cn

**Keywords:** germ cell tumors, histology, age, children, chemotherapy

## Abstract

**Simple Summary:**

A few large series of cases of extracranial germ cell tumors (GCTs) are reported in Asian children. We present a retrospective analysis of extensive scale data of pediatric extracranial GCTs from multiple centers in China. The pathological subtypes and primary sites of tumors showed a preference for occurrence at different ages. For example, sacrococcygeal tumors mostly occur in infants, while mediastinal tumors are common in adolescents. In the context of treatment strategies including platinum-based various chemotherapy regimens, the 5-year overall survival rate and event-free survival rates were 94.13% and 82.33%, respectively. There is no difference in overall survival rate among different chemotherapy regimens for children. The independent clinical factors associated with poor prognosis were that the primary tumor is located in the mediastinum and alpha-fetoprotein levels greater than 10,000 ng/L. In conclusion, the incidence rate and clinical features of extracranial GCTs in children in China are similar to those reported in Europe and the United States. The age distribution of various pathological types and primary sites reflect the characteristics of tumor cell origin of primordial germ cell (PGC) mismigration. For the malignant germ cell tumor with the primary site of mediastinum, more effective treatment regiments should be explored.

**Abstract:**

Objective: The aim was to describe the clinical features of extracranial germ cell tumors (GCTs) in pediatrics and study the clinical risk factors related to survival for malignant germ cell tumors (MGCTs) in order to optimize therapeutic options. Methods: The clinical data of children with extracranial GCTs in three children’s medical centers in Shanghai were retrospectively analyzed. Results: In total, 1007 cases of extracranial GCTs diagnosed between 2010 and 2019 were included in this study, including teratomas (TERs) 706 (70.11%) and MGCTs 301 (29.89%). There were twice as many TER cases as MGCT cases. Approximately 50% of children with GCTs were <3 years old (43.39% for TERs, 67.13% for MGCTs). GCTs in children of different ages show differences in tumor anatomical locations and pathological subtypes. The 5-year event-free survival (EFS) and overall survival (OS) of all patients with MGCTs were 82.33% (95% CI, 77.32%, 86.62%) and 94.13% (95% CI, 90.02%, 96.69%), respectively. The multivariate Cox regression analysis identified a primary site in the mediastinum and alpha fetoprotein (AFP) levels ≥10,000 ng/mL as independent adverse prognostic factors (*p* < 0.0.0001, χ^2^ = 23.6638, *p* = 0.0225, χ^2^ = 5.2072.). There were no significant differences in OS among children receiving various chemotherapy regimens, such as the BEP, PEB, JEB and other regimens (VBP/VIP and AVCP/IEV) (*p* < 0.05). Conclusions: The clinical features of GCTs in Chinese pediatrics are similar to those reported in children in Europe and America. The age distribution of pathological types and primary sites in GCTs reflect the developmental origin of type I and type II GCTs transformed from mismigration primordial germ cells (PGCs). Optimizing the current platinum-based chemotherapy regimens and exploring the treatment strategies for MGCTs of the mediastinum are future research directions.

## 1. Introduction

Extracranial germ cell tumors (GCTs) are rare pediatric cancers that account for approximately 3.5% of all tumors in children under the age of 15 years [1]. However, in adolescents, this proportion increases to 14% [1]. GCTs are a heterogeneous group in age, primary sites, histological features, and prognosis [2]. GCTs arise from primordial germ cells (PGCs) and their derivatives, which evolve into relatively benign teratomas (TERs) and various malignant germ cell tumors (MGCTs) in different environments provided by distinct mismigration pathways [3,4,5], and ultimately there are two common pathological types in pediatrics: type I and type II [6]. Since the 1970s, the survival of patients with MGCTs has dramatically improved with the introduction of cisplatin-based chemotherapy regimens, thereby forming a treatment strategy that combines surgery and chemotherapy [7,8]. There are still some incurable patients despite the application of various platinum-based chemotherapy protocols. More consensus among different international groups on the relative importance of tumor site, tumor stage, age of onset, and elevation in tumor marker levels is needed. The European and American databases and multicenter research in South Africa have reported the clinical prognostic factors [9,10,11]. However, the multicenter data on MGCTs for children in Asia are rarely presented.

We summarized the clinical data of children with extracranial GCTs in three large children’s medical centers in Shanghai in the past decade. These children come from all over the country and can represent the general situation of pediatric GCTs in China. We described the overview of pediatric GCTs and the distribution of onset age, tumor location, and pathological types and analyzed clinical factors related to survival, to reveal the relationship between clinical age distribution characteristics and tumor pathological origin, as well as the adverse prognostic factors of GCTs under the current treatment status.

## 2. Materials and Methods

### 2.1. The Diagnosis and Staging of Patients

The clinical data of patients who were enrolled in Shanghai Children’s Medical Center, Shanghai Children’s Hospital, and Children’s Hospital of Fudan University from January 2010 to December 2019 and diagnosed with extracranial germ cell tumors were reviewed, excluding children with uncertain diagnoses and incomplete records. An imaging examination was performed before the operation, and serum AFP and human chorionic gonadotropin (HCG) levels were measured before treatment. The COG criteria were used for staging at diagnosis [12,13]. In all the centers, the diagnosis of GCTs was mainly based on pathological confirmation, and a few cases were diagnosed according to elevated serum AFP levels and imaging (8 patients with sacrococcygeal tumors). Age at diagnosis, sex, tumor markers, pathology, imaging, and treatment method were analyzed.

### 2.2. Treatment, Monitoring, and Follow-Up

Gonadal primary tumors always required complete surgical organ ablation. Testicular tumors were resected via the inguinal route. Omental and peritoneal lesions were removed with ovarian tumors. Giant MGCTs in the sacrococcygeal region were treated first with neoadjuvant chemotherapy to reduce the tumor size, and then completely removed together with the coccyx.

All patients with MGCTs received chemotherapy except for those with stage I gonadal tumors. Various chemotherapy regimens were adopted by each center, including PEB, JEB, BEP, and AVCP/IEV regimens, as follows:PEB regimen: cisplatin, etoposide, bleomycin (one dose of bleomycin per cycle).JEB regimen: carboplatin, etoposide, bleomycin.BEP regimen: bleomycin, etoposide, cisplatin (3 doses of bleomycin per cycle).AVCP/IEV regimen: AVCP, adriamycin, vincristine, cyclophosphamide, cisplatin; IEV, isocyclophosphamide, etoposide, vincristine.

For MGCTs, the AFP levels were monitored in each chemotherapy cycle during treatment, every two months in the first year after treatment and every three months in the second and third year after treatment. Children with TERs were followed up with imaging every 3 months for 3 years.

### 2.3. Statistical Methods

Statistical analyses were performed and figures were developed using R (version 4.1.2). Overall survival (OS) was defined as the time from diagnosis until death or last patient contact. Event-free survival (EFS) was defined as the time from diagnosis until disease progression, second malignant neoplasm, death, or last patient contact, whichever occurred first. Univariate and multivariate Cox regression analyses were conducted to identify prognostic factors. Survival curves were constructed using Kaplan-Meier and log-rank tests to analyze the differences in OS among the chemotherapy groups. All tests were two-sided, and *p* < 0.05 was considered statistically significant.

### 2.4. Ethical Approval

The study obtained approval as required by the Ethics Review Committee, Shanghai Children’s Hospital, Shanghai Jiao Tong University Approval. protocol code 2023R069-E01 and approval on 7 July 2023).

## 3. Results

### 3.1. Clinical Characteristics and Age Distribution by Pathological Type and Anatomical Location

In the past decade, there have been 1007 newly diagnosed extracranial GCTs in children in the three hospitals, including 706 TERs (70.11%) and 301 MGCTs (29.89%). For TERs, the median age at diagnosis was 68.4 months, and the number of TERs in females was 2.8 times higher than that in males. The highest number of TERs cases were at infancy, while the second peak emerged at the ages of 9–10 years old (Figure 1A). The most common primary site of TERs was the ovary (39.86%, 281), followed by the sacrococcygeal region (20.53%, 145), testis (14.46%, 102), abdomen (mainly from the retroperitoneum, with a few cases of tumors originating from abdominal organs: 13.05%, 92), thorax (mainly from the mediastinum: 7.94%, 56), pelvic cavity (other pelvic areas except for ovarian and typical sacrococcygeal tumors: 2.83%, 20), and others (nasopharynx, neck, hip, femur, and unknown: 1.42%, 10). The pathological type of most patients was mature teratoma (MT, 86.77%), and 12.88% of patients had immature teratoma (IT).

For the 301 MGCTs, the median age at diagnosis was 41 months, and the case numbers were comparable in males and females. The peak incidence was under 3 years old, followed by a steep increase, starting at puberty (Figure 1B). The testicles, sacrococcygeal region, and ovaries were the main primary sites of MGCTs, accounting for 37.21%, 19.93%, and 16.61% of tumors, respectively. Yolk sac tumors (YST) and mixed germ cell tumors are common pathological types of MGCTs, accounting for 63.46% and 30.56%, respectively. The characteristics of the GCTs are summarized in Table 1.

The distribution of the anatomical locations and the histopathological subentities varied according to age at diagnosis and age was divided into several age groups (Figure 2). Whether it was teratoma or MGCTs, approximately 50% of cases were before the age of 3 years. In terms of the primary site of extragonadal GCTs, sacrococcygeal, pelvic, and retroperitoneal tumors mostly occurred before the age of 3 years, while tumors in mediastinal mainly emerged after puberty. Pathological types such as YST, TERs, dysgerminoma/seminoma, and non-seminoma also exhibit age-specificity (Figure 2).

### 3.2. Tumor Markers in MGCTs

Among cases undergoing survival analysis, 246 MGCT were tested for AFP at the initial diagnosis, of which 29 had documented values of “greater than 3000 ng/mL”, making it impossible to accurately determine whether their levels were greater than 10,000 ng/mL. Therefore, an AFP analysis was performed on 217 patients. The Cochran-Mantel-Haenszel test was used to explore the connection between AFP levels and the clinical features of MGCTs. High levels of AFP (>10,000 ng/mL) were common in children with YST and malignant mixed germ cell tumors and were closely associated with stage IV tumors. The AFP levels in patients with dysgerminomas were all normal (Table 2).

### 3.3. Survival and Prognostic Indicators of MGCTs

A total of 279 children with complete follow-up notes were included in the survival analysis. The follow-up time ranged from 1.33 to 13.24 years, and the mean duration of follow-up was 5.3 ± 2.1 years. For the whole cohort of MGCT patients, the 5-year EFS and OS were 82.33% (95% CI, 77.32%, 86.62%) and 94.13% (95% CI, 90.02%, 96.69%), respectively. According to univariate analysis, children with tumors originating in the gonads had significantly better survival than those with tumors originating in the extragonadal region (EFS 91.38 vs. 73.78%, OS 97.70% vs. 88.31%, *p* < 0.01); the survival rate of children with mediastinum tumors was lower than that of children with nonthoracic primary tumors (EFS 41.26% vs. 85.69%, OS 63.07% vs. 95.11%, *p* < 0.01). There was a significant difference in survival between children with complete and partial tumor resection (EFS 88.38% vs. 39.93%, OS 95.59% vs. 76.56%, *p* < 0.01). The EFS and OS for patients with stage I, II, III, and IV MGCTs showed a decreasing trend, with EFS rates of 95.27%, 93.06%, 73.63%, 70.43% (*p* < 0.01) and OS rates of 100%, 93.31%, 87.69%, 84.64% (*p* = 0.04), respectively. There was no significant difference in OS between metastatic and nonmetastatic patients, as well as between patients with different pathological types (YST, mixed germ cell tumors, and other pathological subtypes). There were no significant differences in either EFS or OS between different genders, between patients aged ≥11 years and <11 years, or between patients undergoing tumor resection before and after chemotherapy (Appendix A). The multivariate Cox regression analysis identified primary tumor in the mediastinum and AFP level ˃ 10,000 ng/mL as independent adverse prognostic factors (χ^2^ = 13.4262, *p* < 0.0.01; χ^2^ = 5.2766, *p* = 0.0216).

### 3.4. Chemotherapy for MGCTs

Children with neoplasm stage I Gonadal MGCTs whose postoperative AFP level decreased as expected were managed with observation, and other children with MGCTs underwent various chemotherapy regimens, including BEP (20 patients), PEB (19 patients), JEB (97 patients), and other chemotherapy regimens (AVCP/IEV, VBP/VIP, or VAC/PVB protocol) (99 patients). Three additional chemotherapy courses were administered after the normalization of tumor markers, with total cycles of *n* + 3, except for patients receiving the JEB protocol: patients with stage I–II underwent 4 cycles and patients with stage III–IV underwent 6 cycles. There was no significant difference in OS among these children receiving various chemotherapy regimens (Appendix A).

## 4. Discussion

GCTs are rare neoplasms representing heterogeneity in clinical features. There are few reports on the characteristics and risk factors for GCTs in Asian children. We presented extensive scale data of pediatric extracranial GCTs in China, demonstrating distinct clinical patterns that most likely reflect biological differences.

### 4.1. Prognosis-Related Factors

Platinum-based chemotherapy has significantly improved the outcomes of children with MGCTs. Despite good outcomes overall, the likelihood of a cure for certain sites and histologic conditions is less than 50% [14]. The treatment for patients at a high risk of recurrence still needs to be improved. Several studies have explored the clinical factors conferring a survival disadvantage to pediatric MGCT patients, including an AFP level ˃ 10,000 ng/mL, primary extragonadal tumors, an age greater than 11 years, a clinical stage of IV and residual disease after surgery [15,16,17]. Similar conclusions were obtained, except for children older than 11 years old. It may be that there are fewer children over 14 years old in this data set, as the maximum age allowed for admission to children’s hospitals in Shanghai is usually 14.

In the univariate analysis covering various types of MGCTs in this group, there is no difference in OS between metastatic and non-metastatic cases. Perhaps it is because under the current treatment strategy, even children with metastasis, such as those MGCT originating from the sacrococcygeal region with lung metastasis, will not have poor survival. There is no difference in OS between patients with stage IV and patients with stage I–III, but prognosis of stage IV patients was significantly worse than stage I–II, not stage III, for children with local infiltration (stage III) have poor prognosis, for example, mediastinal MGCT with pleural infiltration (Appendix A). It is meaningful only in homogeneous diseases to compare the survival status between metastatic and nonmetastatic.

Multivariate Cox regression analysis considering the impact of multiple clinical features on survival rate, AFP level ˃ 10,000 ng/mL (discussed below), and a primary tumor of the mediastinum are independent adverse prognostic factors despite prompt cisplatin-based chemotherapy followed by aggressive thoracic surgery. Mediastinal MGCTs confer a survival disadvantage, which is similar to previous reports [16]. A primary mediastinal tumor is a poor prognostic factor in the IGCCC prognosis system for the diagnosis and treatment of adult GCTs [18,19]. In children, there is no consensus or guideline that includes the primary site of the mediastinum as an adverse prognostic factor of MGCTs, and more attention is given to adolescent patients. This may be because most primary mediastinal MGCTs occur in adolescents and young adults [20], so few cases of mediastinal MGCTs have been specifically studied in pediatric clinical practice [21,22]. According to our research and the literature, primary mediastinal tumor should be an important adverse prognostic factor for treatment consideration.

### 4.2. AFP

Clinically, a rise in AFP levels above the age-related normal level is considered elevation. AFP levels are elevated in most children with MGCTs. Our research shows that a high level of AFP is related to the YST and mixed germ cell tumors, high tumor staging, and an AFP level ˃ 10,000 ng/mL is an independent adverse prognostic factor. Previous studies on elevated AFP levels as a worse prognostic factor are not consistent [12,23]. Data from the Children’s Oncology Group(COG) and the Children’s Cancer and Leukemia Group have proven high AFP levels to be a poor prognostic factor [10]. The International Germ Cell Cancer Collaborative Group (IGCCCG) classification has identified an AFP level >10,000 ng/mL as one of the factors determining poor prognosis [24]. As a marker of GCTs, the AFP is an important indicator for diagnosis, tumor burden assessment, and monitoring for recurrence during follow-up. Children with MGCTs show elevated serum AFP levels that are usually 3.5 times the normal upper limit [25]. In our MGCT data, almost all abnormal AFP values were 5 times higher than the upper limit of the institution’s normal value, and a value 5 times higher than the upper limit of the normal is usually an indicator of tumor recurrence [26].

### 4.3. Platinum-Based Chemotherapy Regimen

The treatment strategy for children with MGCTs combines surgery and chemotherapy, which is derived from the adult BEP chemotherapy protocol [27]. Subsequently, the PEB protocol, an improved regimen for children, was developed by COG through a series of clinical trials. Considering the side effects of cisplatin in children, the JEB scheme, in which cisplatin is replaced by carboplatin, has been adopted in the UK and optimized by the clinical trials GCII [21] and GCIII [22]. Other chemotherapy schemes include the VAC or VAC/PVB in the USA [28,29], the VBP/VIP in France [30], and the AVCP/IEV protocol in China. All the above chemotherapy regimens have achieved good survival in their respective applications. We compared the PEB, JEB, and other chemotherapy regimens and found no significant difference in OS, consistent with that of the retrospective comparison of the PEB and JEB protocols by the International Federation of Germ Cell Tumors [31]. A prospective study on which protocol is better (cisplatin or carboplatin) is being conducted by the COG (NCT03067181). The treatment of our patients included various platinum-based chemotherapy regimens and showed high survival rates in all patients. However, there was a significant difference in survival rate in various subgroups. The survival rate of children in stage IV is significantly lower than that of children in stage II, which is even lower in children with mediastinal MGCTs. Further research should focus on reducing the intensity of chemotherapy for low-risk patients and exploring more effective chemotherapy for high-risk patients. The compressed PEB protocol was developed for the treatment of stage II ovarian MGCTs to reduce short-term and long-term side effects and improve quality of life [32,33]. Moreover, it is necessary to explore more effective chemotherapy regimens for patients with clinical risk factors, especially for those with tumors located in the mediastinum.

### 4.4. Age Distribution of GCTs Reflects Pathological Origin

Overall, the onset age is mostly within 3 years of age, and the other onset age peak was in puberty, which is similar to previous reports [34]. The incidence, primary site, and pathological types were shown to be age-related as reported in the literature [35]. YST and mixed germ cell tumors, which may be including YST components, accounted for more than 80% of the total pathological types of MGCTs and were mainly in prepubertal, the vast majority within children under 3 years of age, while all other histological entities such as germinoma/dysgerminomas and choriocarcinoma, were mainly seen in children over 10 years old. These results are also similar to those in the European and American multicenter databases [36].

The age distribution of the primary site and pathological subtypes of GCTs are prominent in our study and are shown in illustration (Figure 2). The anatomic site of primary tumor from bottom to top of body (in the sacrococcygeal region, pelvic cavity, retroperitoneum, and mediastinum) correspond to onset age preference which also being from young children to older children (in newborns, infants, young children, and adolescence), precisely reflecting the process of tumor cell origin (developing from PGC arrested in their migration along the midline of the body during the development of human embryos) [6]. The pathological types I of YST and TERs mainly occur before the age of 6 years, while the pathological types II of dysgerminoma/seminoma and non-seminomatous tumors (NSTs) are more common after puberty, consistent with the new broad classification of GCTs [3]. As we know, at a certain time during embryonic development, the mismigrated PGCs settle in a certain anatomical location of its migration path, encountering diverse niches thus transforming into different GCTs [37]. The distinct age-specific patterns by anatomical site and pathological subclass of GCTs just mirror such complex and development-related aetiology.

Asian children with GCTs exhibit clinical features similar to those of European and American children. However, in terms of the most common primary site of GCTs, our data shows that testicular cases are the most common in MGCTs, while ovarian cases being the most common in TERs, and ovarian cases are also the most common in total GCTs (32.97%), even after removing dermoid cysts from TERs. This is different from the concept that sacrococcygeal GCT is the most commonly formed based on the data from Europe and America. Perhaps there is a difference between Asians and Europeans and Americans [38].The characteristic of distinct age preferences of the primary site and pathological subtypes may be attributed to the unique pathological origin of GCTs tumor cells. Under current treatment strategies, the survival of MGCTs originating from the mediastinum is still poor.

## 5. Conclusions

The clinical characteristics of extracranial GCTs in Asian children are similar to those reported in European and American. The introduction of platinum based chemotherapy regimens has greatly improved the survival rate of children with GCTs. However, some children still have poor prognosis. serum AFP greater than 10,000 ng/mL and MGCT originating from the mediastinum are independent poor prognostic factors, therefore multicenter prospective collaborative research is needed to explore more effective treatment strategies for these patients.

This set of research data lacks cases over the age of 14, so the research results cannot reflect the true status of children with GCTs in this age group. Because it is a retrospective study, some results, such as that there is similar survival of children treated with various chemotherapy schemes, need to be confirmed by a prospective cohort study.

## Figures and Tables

**Figure 1 cancers-15-05412-f001:**
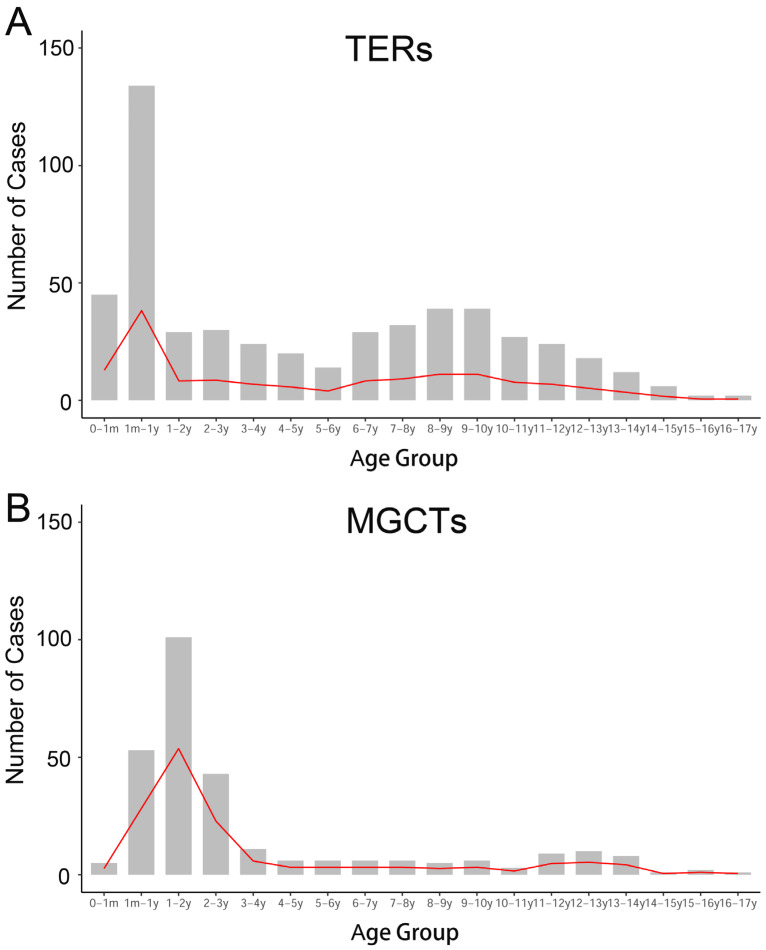
Frequency of occurrence at different ages. (**A**) The early peak of TERs occurred within one year of age, namely the infancy and neonatal period; another peak was during the school age. (**B**) The first peak of MGCTs emerged before the age of three; and a steep increase at puberty.

**Figure 2 cancers-15-05412-f002:**
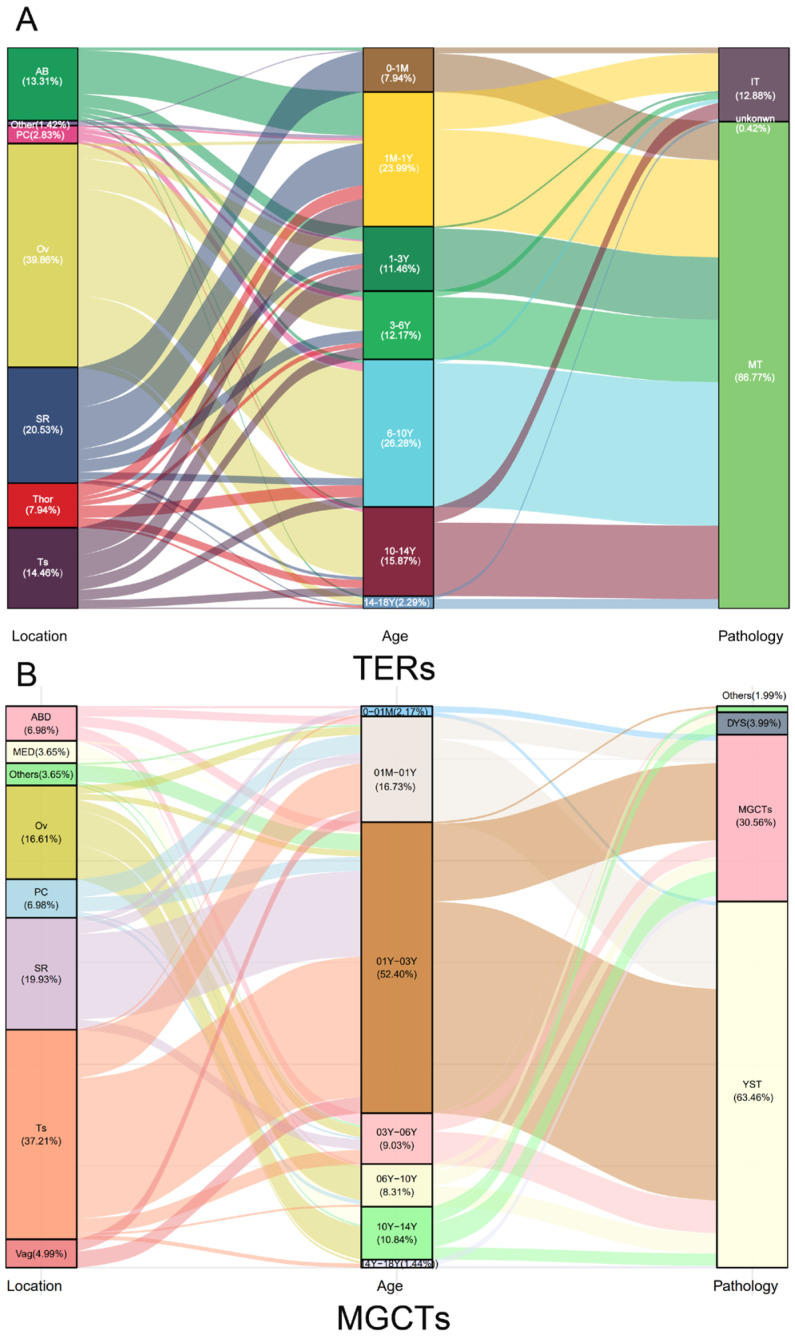
The distribution of GCTs in various children age groups by primary site and pathological subtypes. Generally, the children in pediatrics are divided into several age groups: newborns (0–1 M), infancy (1 M–1 Y), toddler age (1–3 Y), preschool age (3–6 Y), school age (6–10 Y), puberty (10–14 Y), and teenage (14–18 Y). Extragonadal GCTs originating from the sacrococcygeal region, pelvic cavity, vagina, abdomen, and mediastinum, mirroring the migration trajectory of PGC during embryonic development, corresponded to various age at diagnosis which were in the order from newborn to puberty. YST and TERs were dominant before the age of 6 years, whereas dysgerminoma/seminoma and some rare pathological types such as choriocarcinoma and embryonic cancer mainly appeared after puberty. Gonadal GCTs were usually in children after 1 year of age, testicular GCTs mainly occurred from infants to preschoolers, with single tumor component YST or TERs, and ovarian tumors mostly occurred in children from toddler to school age, with dysgerminoma or mixed germ cell tumors or TERs.

**Table 1 cancers-15-05412-t001:** The characteristics of GCTs.

Ters (*n* = 706)		MGCTs (*n* = 301)	
Gender		Gender	
Male/Female	1/2.66	Male/Female	1.13/1
Age (M)	68.40 (3 day–202 month)	Age (M)	41 (3 day–213 month)
Primary site	Proportion (number)	Primary site	Proportion (number)
ovary	39.86% (281)	testis	37.21%(112)
sacrococcygeal	20.53% (145)	ovary	16.61% (50)
testis	14.46% (102)	sacrococcygeal	19.93% (60)
abdomen	13.05% (92)	abdomen	6.98% (21)
pelvic cavity	2.83% (20)	pelvic cavity	6.98% (21)
thorax	7.94% (56)	Thorax(mediastinum)	3.65% (11)
Others (nasopharynx, neck, hip, thymus, femur)	0.008% (6)	Vagina	4.99% (15)
unknown	0.005% (4)	Others (orbit, upper lip, etc.)	2.66% (8)
Pathology	Proportion (number)	unknown	0.10% (3)
Mature teratoma	86.77% (612)		
Immature teratoma	12.88% (91)	Pathology	
Unknown	0.42% (3)	Yolk sac tumor	63.45% (191)
		Mixed germ cell tumor	30.56% (92)
		dysgerminoma	3.99% (12)
		gonadoblastoma	0.10% (3)
		seminoma	0.33% (1)
		Embryonal carcinoma	0.33% (1)
		Choriocarcinoma	0.33% (1)
		Staging	
		I	37.21 (112)
		II	11.63 (35)
		III	27.57 (83)
		IV	23.59 (71)

**Table 2 cancers-15-05412-t002:** Correlation of AFP in MGCTs with pathology and staging.

	N	AFP (No./%)				CMH Statistics	*p*
		S0	S1	S2	S3		
Total	217	24/11.06	37/17.05	66/30.41	91/41.94		
Pathology						81.32	<0.0001
Yolk sac tumor	137	4/2.92	25/18.25	53/38.69	55/40.15		
Mixed germ cell tumour	67	8/11.94	11/16.42	12/17.91	36/53.73		
Choriocarcinoma	1	1	0	0	0		
Embryonal carcinoma	1	0	0	1	0		
Dysgerminoma	11	11/4.72	0	0	0		
Seminoma	1	1/0.43	0	0	0		
Staging						33.21	<0.0001
Ⅰ	70	6/8.57	22/31.43	33/47.14	9/12.86		
Ⅱ	25	7/28.00	6/24.00	8/32.00	6/24.00		
Ⅲ	59	9/15.25	4/6.78	18/30.51	32/54.24		
Ⅳ	58	2/3.45	5/13.79	6/10.34	45/77.59		

S0: within normal reference values, S1: above age-related normal values and <1000 ng/mL, S2: 1000–10,000 ng/mL, S3: >10,000 ng/mL.

## Data Availability

Restrictions apply to the availability of these data. Data was obtained from Three Children’s Medical Centers in Shanghai and are available from Case systems in various hospitals with the consent of each hospital.

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
