# Peer review of "Extracranial Germ Cell Tumors in Children: Ten Years of Experience in Three Children’s Medical Centers in Shanghai"

_cancers, 2023, doi:10.3390/cancers15225412_

Round 1
Reviewer 1 Report (Previous Reviewer 3)
Comments and Suggestions for Authors
See attached file

Comments on the Quality of English Language
Most of the comments are textual.
Author Response
please see the attachment.

Reviewer 2 Report (Previous Reviewer 2)
Comments and Suggestions for Authors
Thank you for your explanations.
Comments on the Quality of English Language
Please double check the English language.
Author Response
plesse see the attachment.

This manuscript is a resubmission of an earlier submission. The following is a list of the peer review reports and author responses from that submission.
Round 1
Reviewer 1 Report
Comments and Suggestions for Authors
Many thanks for asking me to review this retrospective review of pediatric germ cell tumours from three medical centers in Shanghai, China over a ten-year period.
The paper requires a major revision before it can be considered for publication. My comments below relate and hopefully provide helpful commentary to develop the manuscript.
I wish the authors all the best with the revision which I hope they will undertake.
Points to address:
General form
1. Please could the methods section appear after the introduction as is customary in most journals. It seems odd to leave it until after the discussion.
Introduction
2. The South African study mentioned in the introduction in line 49 is not referenced. Please add the correct reference. Also reference 21 (Frazier et al for the MaGIC group is probably a better fit there).
Methods
3. Please be specific about how many diagnoses were pathologically confirmed and how many were based on elevated markers and imaging (page 9, lines 296-297).
4. Could you explain why ovarian stage 1 patients received chemotherapy? (page 10, Line 304).
5. Does your local ethics committees issue ethical approval numbers? I would mention the specific names of these committees and/or provide the ethical approval numbers attached to the approvals. (page 10, line 326)
Results
6. Would a thymic location not be considered mediastinal/thorax? (page 2, line 67)
7. What proportion of the patients with ITs had elevated tumour markers at diagnosis outside of the reference range for age? These may in fact be considered to be mixed germ cell tumours and should then be included in the analysis with MGCTs. As per the latest WHO classification any child with an IT and AFP > 1000ng/ml at diagnosis or any percentage YST elements is considered a mixed GCT and should be treated as such. (Page 2, line 69)
8. TABLE 1 and Page 5, Line 111 and page 6 line 128. Sex cord stromal tumours are not germ cell tumours and should be removed from the analysis. They are derived from stromal, sustentacular tissue adjacent to germ cells not from germ cells themselves.
9. Page 5, Line 112: Please replace embryonic cancer with embryonal carcinoma if that is what is meant consistent with TABLE 1.
10. TABLE 2 and page 6, line 138. Please define low-, medium and high risk groups in your methods section.
11. Page 6, line 137: The number of patients with very elevated values are higher in the stage 3 patients than stage 4 patients according to TABLE 2 but you make no mention of that. Can I suggest that you combine stage 3 and 4 patients (advanced disease) and stage 1 and 2 patients to make TABLE 2 more rational. The ‘TOTALS’ column is not useful in the stage section of TABLE 2 as the 79 stage 1 patients had much lower AFP elevations than those in the stage 4 group. Can you see how making a ‘TOTALS’ column gives the impression that the numbers in limited and advanced stage patients is almost the same?
Discussion
12. Any discussion should begin with the main findings of the study.
13. Page 7 and 8, lines 195-222 are really more like a review that should not appear here. If anything it can shortened and made part of the introduction.
14. Reference 24, page 8, line 238. This is a paper is about reduced and compressed cisplatin in IR GCTs in children and adolescents which found that 3 courses of PEb to have inferior EFS at a set target and recommends 4. Some sub-groups had excellent outcomes but the study was insufficiently powered to make comparisons between sub-groups. Nowhere in this paper can I find evidence to support the statement that AFP values >5X normal is an indicator of tumour recurrence. I am happy to be corrected or can I suggest that a more appropriate reference is used.
15. Section 3.3 and 3.4: Can I suggest that you start with your findings and then compare and contrast to other studies but avoid doing a review. That is not the purpose of a discussion.
16. Please include a section on the limitations of your study.
Conclusion
17. This should be a focussed summary which answers the question, “Why did you undertake this study?”. I would also think about what the implications are for your practice going forward.
Comments on the Quality of English Language
Minor edits required.
Author Response
Response to Reviewer 1 Comments
Point 1: Please could the methods section appear after the introduction as is customary in most journals. It seems odd to leave it until after the discussion.
Response 1: The format of this article is based on similar literature from 'Cancer' magazine (Cancer 2020, 12, 611; doi: 10.3390/Cancer 12030611). As customary in most journals, the method section has been placed after the introduction in the paper.
Point 2: The South African study mentioned in the introduction in line 49 is not referenced. Please add the correct reference. Also reference 21 (Frazier et al for the MaGIC group is probably a better fit there).
Response 2: Good suggestion, this reference "21" has been cited in the introduction.
Point 3: Please be specific about how many diagnoses were pathologically confirmed and how many were based on elevated markers and imaging (page 9, lines 296-297).
Response 3: In this set of case data,eight cases of sacrococcygeal Germ cell tumor were clinically diagnosed as Germ cell tumor on the basis of imaging and alpha fetoprotein elevation.
Point 4: Could you explain why ovarian stage 1 patients received chemotherapy? (page 10, Line 304).
Response 4: This is a clerical error. Actually, it should be for phase I MGCT of gonads, without the need for chemotherapy.
Point 5: Does your local ethics committees issue ethical approval numbers? I would mention the specific names of these committees and/or provide the ethical approval numbers attached to the approvals. (page 10, line 326).
Response 5: the Ethics Review Committee, Children's Hospital of Shanghai/Shanghai Children’s Hospital, Shanghai Jiao Tong University Approval. protocol code 2023R069-E01 and approval on July 7th, 2023.
Point 6: Would a thymic location not be considered mediastinal/thorax? (page 2, line 67).
Respons 6: Yes, a considerable portion of chest tumors in this group originated from the thymus gland.
Point 7: What proportion of the patients with ITs had elevated tumour markers at diagnosis outside of the reference range for age? These may in fact be considered to be mixed germ cell tumours and should then be included in the analysis with MGCTs. As per the latest WHO classification any child with an IT and AFP > 1000ng/ml at diagnosis or any percentage YST elements is considered a mixed GCT and should be treated as such. (Page 2, line 69).
Respons 7: You're right. We have clinically diagnosed a case of neonatal sacrococcygeal mature Teratoma with high serum alpha fetoprotein. One year after surgical resection of the tumor, the child was diagnosed as MGCT.
This article focuses on the analysis of MGCTs, and has not yet paid attention to the alpha fetoprotein of mature Teratoma. My colleague may be about to conduct a study on Teratoma.
Point 8: TABLE 1 and Page 5, Line 111 and page 6 line 128. Sex cord stromal tumours are not germ cell tumours and should be removed from the analysis. They are derived from stromal, sustentacular tissue adjacent to germ cells not from germ cells themselves.
Respons 8: It should be mixed Germ cell sex cord stromal tumor. After careful review, it was found that there were indeed cases of sex cord stromal tumors included. We are preparing to exclude these cases and reanalyze them.
Point 9: Page 5, Line 112: Please replace embryonic cancer with embryonal carcinoma if that is what is meant consistent with TABLE 1.
Respons 9: The embryonic cancer has been replaced with embryonal carcinoma in the manuscript.
Point 10: TABLE 2 and page 6, line 138. Please define low-, medium and high risk groups in your methods section.
Respons 10: Considering the inconsistent definitions of low risk, medium risk, and high-risk populations among the three children’s medical centers, we have removed the analysis section for these risk groups from Table 2.
Point 11: Page 6, line 137: The number of patients with very elevated values are higher in the stage 3 patients than stage 4 patients according to TABLE 2 but you make no mention of that. Can I suggest that you combine stage 3 and 4 patients (advanced disease) and stage 1 and 2 patients to make TABLE 2 more rational. The ‘TOTALS’ column is not useful in the stage section of TABLE 2 as the 79 stage 1 patients had much lower AFP elevations than those in the stage 4 group. Can you see how making a ‘TOTALS’ column gives the impression that the numbers in limited and advanced stage patients is almost the same?
Respons 11: Okay, we will refer to your proposal to improve Table 2.
Point 12: Any discussion should begin with the main findings of the study.
Respons 12: Okay, we will refine the discussion section.
Point 13: Page 7 and 8, lines 195-222 are really more like a review that should not appear here. If anything it can shortened and made part of the introduction.
Respons 13: Here we want to explain that clinically, the primary sites of Germ cell tumor occurs from the bottom to the top (sacrococcygeal, pelvic, mediastinum) distributed in in neonates, young children and older children, which corresponds to the gradual evolution of primitive Germ cell into tumor cells in the process of mismigration from bottom to top of body. We have improved the discussion on sections 3.1 and 3.2. We consider that although the characteristics of age distribution are not a first found of us, but this paper shows this feature intuitively by figure 2 and explains it reasonably from the perspective of tumor cell origin for the first time.
Point 14: Reference 24, page 8, line 238. This is a paper is about reduced and compressed cisplatin in IR GCTs in children and adolescents which found that 3 courses of PEb to have inferior EFS at a set target and recommends 4. Some sub-groups had excellent outcomes but the study was insufficiently powered to make comparisons between sub-groups. Nowhere in this paper can I find evidence to support the statement that AFP values >5X normal is an indicator of tumour recurrence. I am happy to be corrected or can I suggest that a more appropriate reference is used.
Respons 14:The original text is “A complete response was defifined as normalization of tumor markers (to within fifive times the upper limit of the institutional normal, to allow for a low-level plateau)” which is in study design line 15.Interestingly, while writing this paper, I remembered reading this description, but couldn't remember which literature it was in. It took me two days to find it.I would be happy if you could suggest a more suitable reference.
Point 15: Section 3.3 and 3.4: Can I suggest that you start with your findings and then compare and contrast to other studies but avoid doing a review. That is not the purpose of a discussion.
Respons 15:Okay, I will try to modify the discussion section according to your suggestion. Anyway, it will take another 2 weeks for our statistical experts to recalculate (as some cases have been deleted, all data needs to be recalculated), and I will further refine this paper in 2 weeks.
Point 16: Please include a section on the limitations of your study.
Respons 16:All right.
Point 17: This should be a focussed summary which answers the question, “Why did you undertake this study?”. I would also think about what the implications are for your practice going forward.
Respons 17:Thank you very much for your guidance. Recent years, our clinical practice has paid particular attention to the treatment of mediastinal Germ cell tumors, which is also the main content of our next paper.
Reviewer 2 Report
Comments and Suggestions for Authors
The study of Jiang et al. is a retrospective multicentre study of three children's medical centres in Shanghai. The study describes clinical features and risk factors in a cohort of 1028 newly diagnosed patients with an extracranial GCT. It is a well described cohort which contributes to the already existing data in the literature.
Methods/Results:
How were the children identified form the different hospitals? Is a (national) registry available for children with GCTs? Could adolescent patients have been missed in case they were referred to specialists for adult patients?
A sex cord stromal tumour is not a germ cell tumour. Sex cord stromal tumours were included in the cohort and analyses. I would suggest to leave them out this paper.
p6, line 138: it is not likely that a sex cord stromal tumour or a dysgerminoma produces AFP.
p7, line158: the statement: 'There was no significant difference in OS between metastatic and nonmetastatic patients...' is confusing to me, since EFS and OS showed a decrease when stage increased from I to IV.
Was an increase in EFS or OS observed during time?
Author Response
Response to Reviewer 2 Comments
Point 1: How were the children identified form the different hospitals? Is a (national) registry available for children with GCTs? Could adolescent patients have been missed in case they were referred to specialists for adult patients?
Response 1: The participating authors of each hospital are responsible for collecting data on newly diagnosed patients with extracranial GCTs from the hospital's case registration system between January 1, 2010 and December 30, 2019, and then submitting these data to Dr. Jiang and Dr. Gao for summary. Most children's hospitals in China only accept children under the age of 14. Children diagnosed in children's hospitals can still be followed up at children's hospitals until they are over 18 years old, even if they are over age during follow-up.
Point 2: A sex cord stromal tumour is not a germ cell tumour. Sex cord stromal tumours were included in the cohort and analyses. I would suggest to leave them out this paper.
Response 2: Thank you for your suggestion. After careful inspection, we found that 12 cases of sex cord stromal tumour were included in the analysis. We have removed these 12 cases data from the study and conducted a new statistical analysis of MGCTs cases. Therefore, the corresponding statistical data, charts in the manuscript , and tables in the supplementary materials have also been modified and the research results and conclusions have not changed as a result.Please refer to the manuscript at lines 32-40, 124-126, 138-145, 171-173, 178-179, 182-184, 204, 207, 218-222, 226-233, 321-323, as well as Figures 1 (147), figure 2 (188), table1 (153), and table 2 (214).
Point 3: p6, line 138: it is not likely that a sex cord stromal tumour or a dysgerminoma produces AFP.
Response 3: The content of the paper has been modified accordingly. "Sex cord stromal tumor" should be mixed Germ cell sex cord stromal tumor, please refer to line 212 of the manuscript.
Point 4: p7, line158: the statement: 'There was no significant difference in OS between metastatic and nonmetastatic patients...' is confusing to me, since EFS and OS showed a decrease when stage increased from I to IV.Was an increase in EFS or OS observed during time?
Response 4: After careful verification, the metastase group and stage IV group are the same group of cases in this analysis data. What's different is that which compared to the group of stage IV group are the stage 1, stage 2, and stage 3 respectively, while which to compared with metastases group is no-metastase group including all children in stage 1 to stage 3, while older children in stage 3 with primary mediastinum and abdomen may have poor prognosis.
We consider that there is no difference in OS between metastatic and non metastatic cases, while the prognosis of stage IV patients is significantly worse because univariate analysis results are influenced by multiple factors such as tumor location, age, and pathological type. We have provided a brief explanation in the manuscript, please refer to lines 342-348 of the manuscript.
Reviewer 3 Report
Comments and Suggestions for Authors
The authors retrospectively describe the clinical features, and the risk factors for treatment failure in the so far largest group of Asian pediatric germ cell tumor (GCT) patients, recruited from three pediatric hospitals in Shanghai. The significance of the study lies in the Asian descent and the size of the patient population. They conclude that their patients resemble those published in European and American studies, including their risk factors for treatment failure.
The maximum age for admission in the three Shanghai pediatric hospitals from which the patients have been recruited is 14 years of age. The authors should explain why, nevertheless, the age range of the patients is less than 3 days through 213 month (close to 18 years).
In this study the most frequent site of type I teratomas is the ovary. This is in disagreement with the literature where the most frequent site consistently is the sacrococcygeal region. Could it be that some of the ovarian teratomas were in fact dermoid cysts (type IV GCTs), which have been classified as mature teratomas?
The anatomical localisations: abdominal, thoracic and pelvic should better defined/specified.
Throughout the manuscript the authors refer to sex cord stromal tumors, which are not GCTs. They most likely mean mixed germ cell sex cord stromal tumors, which are, however, only once mentioned in the paper.
A relative weakness of the study is that the histology of slides was not reviewed and reclassified according to GCT types 0 through VI.

Comments on the Quality of English Language
Suggestions for improvements, among others of the English, are to be found in the attached PDF.
Author Response
Response to Reviewer 3 Comments
Point 1: The maximum age for admission in the three Shanghai pediatric hospitals from which the patients have been recruited is 14 years of age. The authors should explain why, nevertheless, the age range of the patients is less than 3 days through 213 month (close to 18 years).
Response 1: It is not a mandatory requirement for children's hospitals to limit the age of admission to children under 14 years old. Sometimes, if parents of children trust a doctor at the children's hospital more, they can also choose to seek medical treatment at the children's hospital.
Point 2: In this study the most frequent site of type I teratomas is the ovary. This is in disagreement with the literature where the most frequent site consistently is the sacrococcygeal region. Could it be that some of the ovarian teratomas were in fact dermoid cysts (type IV GCTs), which have been classified as mature teratomas?
Response 2: Dermoid cysts were classified as mature Teratoma in this study. In fact, there are only 10 cases of dermoid cysts in this group of data. After removing these 10 cases for further analysis, the ovary remains the most common site . The reasons will be left for my colleague to explore, as she is going to analyzing the data of this group of teratoma cases.
Point 4: Throughout the manuscript the authors refer to sex cord stromal tumors, which are not GCTs. They most likely mean mixed germ cell sex cord stromal tumors, which are, however, only once mentioned in the paper.
Response4: The cases of mixed sex cord stromal tumors in this article include some sex cord stromal tumors. After checking with the original data, 12 cases of sex cord stromal tumors have been excluded from this study and the MGCT case data has been re statistically analyzed. The data and charts in the manuscript have been modified accordingly.
Point 5: A relative weakness of the study is that the histology of slides was not reviewed and reclassified according to GCT types 0 through VI.
Response5: I completely agree with you. However, there are not many doctors in China who pay attention to the detailed staging of germ cell tumors.
Round 2
Reviewer 1 Report
Comments and Suggestions for Authors
There are still some things which are not done e.g. the South African study to which they refer is still not referenced. In point No. 14 the author asks me to provide a reference. They have used reference 24 to cite an AFP threshold (5X normal) but in fact it is a reference inside a reference. That information belongs to two other papers from Morris MJ et al 2000 and Mann JR et al 2000 (references 17 and 18 inside the dose compressed PEb paper).
Comments on the Quality of English Language
Minor editing is needed.
Author Response
Response to Reviewer 1 Comments (Round 2)
Point 1: Please could the methods section appear after the introduction as is customary in most journals. It seems odd to leave it until after the discussion.
Response 1: The method section has been placed after the introduction, please refer to line 76 of the manuscript.
Point 2: The South African study mentioned in the introduction in line 49 is not referenced. Please add the correct reference. Also reference 21 (Frazier et al for the MaGIC group is probably a better fit there).
Response 2: The reference "21" you recommended has become the current reference "11", please refer to line 66 of the manuscript. The word 'South Africa' has been removed from the paper.
Point 3: Please be specific about how many diagnoses were pathologically confirmed and how many were based on elevated markers and imaging (page 9, lines 296-297).
Response 3: In this set of case data,eight cases of sacrococcygeal Germ cell tumor were clinically diagnosed as Germ cell tumor on the basis of imaging and alpha fetoprotein elevation, please refer to line 84 of the manuscript.
Point 4: Could you explain why ovarian stage 1 patients received chemotherapy? (page 10, Line 304).
Response 4: This is a clerical error. Actually, it should be for phase I MGCT of gonads, without the need for chemotherapy. Please refer to lines 93-94 of the manuscript.
Point 5: Does your local ethics committees issue ethical approval numbers? I would mention the specific names of these committees and/or provide the ethical approval numbers attached to the approvals. (page 10, line 326).
Response 5: The Ethics Review Committee, Children's Hospital of Shanghai/Shanghai Children’s Hospital, Shanghai Jiao Tong University Approval. protocol code 2023R069-E01 and approval on July 7th, 2023. Please refer to lines 116-118 of the manuscript.
Point 6: Would a thymic location not be considered mediastinal/thorax? (page 2, line 67).
Respons 6: 'thymus' has been removed from “others”. Please refer to line 133 of the manuscript.
Point 7: What proportion of the patients with ITs had elevated tumour markers at diagnosis outside of the reference range for age? These may in fact be considered to be mixed germ cell tumours and should then be included in the analysis with MGCTs. As per the latest WHO classification any child with an IT and AFP > 1000ng/ml at diagnosis or any percentage YST elements is considered a mixed GCT and should be treated as such. (Page 2, line 69).
Respons 7: This paper mainly analyzes MGCTs. My colleague will be responsible for analyzing and writing a paper on the teratoma. I have conveyed your suggestion to her.
You're right. We have encountered teratoma with elevated afp in clinical practice, and within one year after surgery, mixed germ cell tumors were diagnosed at the same anatomical site.
Point 8: TABLE 1 and Page 5, Line 111 and page 6 line 128. Sex cord stromal tumours are not germ cell tumours and should be removed from the analysis. They are derived from stromal, sustentacular tissue adjacent to germ cells not from germ cells themselves.
Respons 8: In this set of data, 12 cases sex cord stromal tumors were included in the mixed germ cell cord stromal tumors. They have been removed from the MGCTs data now. Table 1, Table 2 and figure2B in the manuscript have been corrected accordingly. Please refer to lines 181 and 200 of the manuscript.
Point 9: Page 5, Line 112: Please replace embryonic cancer with embryonal carcinoma if that is what is meant consistent with TABLE 1.
Respons 9: The embryonic cancer has been replaced with embryonal carcinoma in the manuscript.Please refer to line 182 of the manuscript
Point 10: TABLE 2 and page 6, line 138. Please define low-, medium and high risk groups in your methods section.
Respons 10: Considering the inconsistent definitions of low risk, medium risk, and high-risk populations among the three children’s medical centers, we have removed the analysis section for these risk groups from Table 2, please refer to Line 212, Table 2.
Point 11: Page 6, line 137: The number of patients with very elevated values are higher in the stage 3 patients than stage 4 patients according to TABLE 2 but you make no mention of that. Can I suggest that you combine stage 3 and 4 patients (advanced disease) and stage 1 and 2 patients to make TABLE 2 more rational. The ‘TOTALS’ column is not useful in the stage section of TABLE 2 as the 79 stage 1 patients had much lower AFP elevations than those in the stage 4 group. Can you see how making a ‘TOTALS’ column gives the impression that the numbers in limited and advanced stage patients is almost the same?
Respons 11: Table 2 has been modified according to your suggestion.Please refer to Line 212, Table 2.
Stage 3 MGCTs are mainly seen in older children with mediastinal or ovarian tumors accompanied by local infiltration. This set of materials (Gonadal and extragonadal germ cell tumors )are being analyzed to write manuscript by two of my colleagues respectively.
Point 12: Any discussion should begin with the main findings of the study.
Respons 12: Okay, we will refine the discussion section.
Point 13: Page 7 and 8, lines 195-222 are really more like a review that should not appear here. If anything it can shortened and made part of the introduction.
Respons 13: We analyzed the possible correlation between the age distribution characteristics of germ cell tumors and the pathological origin of tumor cells in this section. This part of the manuscript has been condensed. Please refer to lines 275 to 286 of the manuscript.
Point 14: Reference 24, page 8, line 238. This is a paper is about reduced and compressed cisplatin in IR GCTs in children and adolescents which found that 3 courses of PEb to have inferior EFS at a set target and recommends 4. Some sub-groups had excellent outcomes but the study was insufficiently powered to make comparisons between sub-groups. Nowhere in this paper can I find evidence to support the statement that AFP values >5X normal is an indicator of tumour recurrence. I am happy to be corrected or can I suggest that a more appropriate reference is used.
Respons 14: This two articles “Morris MJ, Bosl GJ: Recognizing abnormal marker results that do not reflflect disease in patients with germ cell tumors. J Urol 163:796-801, 2000” has been cited as a reference” and”Mann JR, Raafat F, Robinson K, et al: The United Kingdom Children’s Cancer Study Group’s second germ cell tumor study: Carboplatin, etoposide, and bleomycin are effective treatment for children with malignant extracranial germ cell tumors, with acceptable toxicity. J Clin Oncol 18:3809-3818, 2000” will be cited as references after being read by us. We have gone to the university library to search for the original text, which will take time.
Point 15: Section 3.3 and 3.4: Can I suggest that you start with your findings and then compare and contrast to other studies but avoid doing a review. That is not the purpose of a discussion.
Respons 15:Yes, there were too many references in our discussion, and this part has been compressed.
The description of clinical characteristics is also the main content of this paper. One of the purposes is to compare whether Chinese children with GCT have similar clinical manifestations to European and American children. The second purpose is to explain the age distribution characteristics of GCTs tumor anatomical sites and pathological types in clinical practice from the origin of GCTs tumor cells. Therefore, the discussion starts with clinical features.
Please refer to lines 260-286 of the manuscript.
Point 16: Please include a section on the limitations of your study.
Respons 16: The limitations of this study are in the last paragraph of the manuscript. Please refer to lines 362-366 of the manuscript.
Point 17: This should be a focussed summary which answers the question, “Why did you undertake this study?”. I would also think about what the implications are for your practice going forward.
Respons 17: Thank you very much for your suggestion. Please refer to lines 356-361 of the manuscript.
Reviewer 3 Report
Comments and Suggestions for Authors
The authors have addressed most questions about the content, like whether there were dermoid cysts among the ovarian teratomas. However, in the process of rewriting the manuscript English language has somewhat deteriorated.
In the supplementary file the name of 'malignant mixed germ tumor cell sex cord stromal tumors' has an error. It should be written as: 'malignant germ cell sex cord stromal tumor'.
My advise is that they correct the manuscript according to the attached file, and have the English checked by a native speaker.

Comments on the Quality of English Language
English language should be checked by a native speaker.
Author Response
Response to Reviewer 3 Comments(round 2)
Point 2: In this study the most frequent site of type I teratomas is the ovary. This is in disagreement with the literature where the most frequent site consistently is the sacrococcygeal region. Could it be that some of the ovarian teratomas were in fact dermoid cysts (type IV GCTs), which have been classified as mature teratomas?
Response 2: Dermoid cysts were classified as mature Teratoma in this study. In fact, there are only 10 cases of dermoid cysts in this group of data. After removing these 10 cases for further analysis, the ovary remains the most common site . The reasons will be left for my colleague to explore, as she is going to analyzing the data of this group of teratoma cases.
Point 4: Throughout the manuscript the authors refer to sex cord stromal tumors, which are not GCTs. They most likely mean mixed germ cell sex cord stromal tumors, which are, however, only once mentioned in the paper.
Response4: The cases of mixed sex cord stromal tumors in this article include some sex cord stromal tumors. After checking with the original data, 12 cases of sex cord stromal tumors have been excluded from this study and the MGCT case data has been re statistically analyzed. The data and charts in the manuscript have been modified accordingly.
Point 5: A relative weakness of the study is that the histology of slides was not reviewed and reclassified according to GCT types 0 through VI.
Response5: I completely agree with you. However, there are not many doctors in China who pay attention to the detailed staging of germ cell tumors.